# Nitrogen Application Can Be Reduced without Affecting Carotenoid Content, Maturation, Shelf Life and Yield of Greenhouse Tomatoes

**DOI:** 10.3390/plants12071553

**Published:** 2023-04-04

**Authors:** Dimitra Chormova, Victor Kavvadias, Edward Okello, Robert Shiel, Kirsten Brandt

**Affiliations:** 1School of Agriculture, Food and Rural Development, Newcastle University, Agriculture Building, Newcastle upon Tyne NE1 7RU, UK; chormova@hotmail.com (D.C.); edward.okello@ncl.ac.uk (E.O.); r.s.shiel@ncl.ac.uk (R.S.); 2Department of Soil Science of Athens, Institute of Soil and Water Resources, Hellenic Agricultural Organization DIMITRA, 1 Sofokli Venizelou Str., Lykovrysi-Athens, 14123 Attiki, Greece; 3Population Health Sciences Institute, Newcastle University, Framlington Place, Newcastle upon Tyne NE2 4HH, UK; kirsten.brandt@newcastle.ac.uk

**Keywords:** *Solanum lycopersicum*, fertiliser, ripening, fruit quality, fruit storage ability

## Abstract

Tomatoes (*Solanum lycopersicum* L.) of the variety Elpida were grown under standard Mediterranean greenhouse conditions during the spring season at three different nitrogen levels (low 6.4, standard 12.8, high 25.9 mM/plant), which were replicated during two consecutive years. Application of high nitrogen significantly increased the colour index a* (*p* < 0.001) but did not significantly affect yield or quality. The variety exhibited prolonged postharvest storage at room temperature (median survival time of 93 days). The maturation process was delayed by harvest at the breaker stage (2.5 days, *p* ≤ 0.001) or by super-optimal temperatures in the second year of experimentation (10 days, *p* ≤ 0.001). The colour indices L* and a* and the hue angle (a/b*) were positively correlated with the sum of total carotenoids, while differences in b* depended on the year of cultivation. The sustainability of this type of tomato production can be improved by reducing the nitrogen supply to less than the current standard practice, with minimal risk or negative effects on yield and quality of tomatoes.

## 1. Introduction

Several prospective epidemiological studies suggest that the consumption of tomato (*Solanum lycopersicum* L.) could decrease the risk of cardiovascular disease and certain forms of cancer [1,2]. While previous hypotheses focusing on lycopene and other carotenoids [3,4,5] were not supported by experimental evidence [6,7], intervention studies with raw tomato still indicate benefits for cardiovascular health markers [8,9], justifying a continued focus on increasing the intake of raw tomato. Plant nutrition management has an important effect on the growth and yield of plants [10] and can influence the concentrations of compounds that influence consumer preference, such as carotenoids [11,12]. However, other factors such as light and temperature can also affect growth and development and influence the synthesis of these compounds [13,14]. 

Yield plays a vital economic role for the growers, and sufficient fertilization is necessary to achieve profitable results [15]. According to Massey and Winsor [16], the production of tomato fruit can be achieved by providing a wide range of nitrogen concentrations, from 0.7 to 23 mM. A number of experimental studies have investigated the effect of different nitrogen levels on the yield of greenhouse tomato in the Mediterranean region and reported that standard doses (7–14 mM) resulted in the greatest yield [17,18,19] or that there was no significant effect of the different nitrogen treatments on the yield of marketable fruits [20,21]. Regarding quality, Bénard et al. [22] reported that lowering nitrogen supply from 12 to either 6 or 4 mM NO_3_^−^ had a minimal impact on commercial yield (7.5%), but it increased fruit dry matter content, consequently improving fruit quality due to lower acid content (10−16%) and increased soluble sugar content (5–17%).

Although the main vegetable crops in Greek greenhouses are tomatoes and cucumbers, followed by peppers [23], information in Greece regarding optimal nutrient management for greenhouse tomato, and particularly in relation to the application of nitrogen, is insufficient and does not provide useful guidelines that are relevant and applicable to the growers. Until now, no studies have reported any information related to the postharvest or preharvest maturation of tomatoes, taking into consideration the shelf life of a commercial variety.

Maturation is the process from flower development until tomato fruits reach the red stage. A fruit can be mature in size but not ripe [24]. Ripening is the process from the breaker to the red stage, and it is mainly attributed to the ethylene released by the fruits [25,26]. The tomato is classified as a climacteric fruit. In fact, the ripening of climacteric fruits is accompanied by a distinct increase in respiratory rate due to increased ethylene production just before the increase in respiration. After the increase in respiration, which results in the maturation of the fruit, ethylene production declines significantly [27]. Tomato fruits can ripen on the plants (on-vine) or be removed at the turning or pink stage and ripen in postharvest conditions (off-vine) [28]. Agricultural techniques such as fertilizer applications or cultivation systems can affect the maturation of fruits and vegetables [10,29,30]. Moreover, the use of beneficial soil microorganisms could be an effective strategy for alleviating or minimizing the negative effects of climatic conditions on plant growth [15].

Tomato fruits contain several carotenoids, with lycopene being the major component. It is generally observed that relatively high nitrogen doses increase the carotenoid content of fruits and vegetables [31]. Chenard et al. [32] observed that nitrogen levels two and four times above the standard dose (1.8 mM) compared to low and standard levels resulted in higher carotenoid concentrations. Similar results were observed by other experimental studies [33,34]. However, other studies reported no significant effects of different nitrogen doses on the carotenoid content of fruits and vegetables [22,35,36,37,38]. Furthermore, excessive nitrogen supply had a negative impact on tomato attributes [39].

Colour indices L*, a* and b* are used to measure the colour intensity and hue of fruits and vegetables. Fruits that develop a uniformly intense colour are more appealing to consumers. Arias et al. [40] and Akbudak et al. [41] found positive correlations between a* (red hue) and the lycopene content of tomato fruits at different ripening stages. A positive effect of nitrogen fertilization on the red colour of tomato fruit, related to the increment of lycopene concentration, has also been recorded [40,42]. Applications of nitrogen levels five times above the standard dose increased the colour index a* compared to standard and unfertilized treatments [37]. Increasing nitrogen dose resulted in an increase in a* and the a*/b* ratio, while L* (lightness) decreased [43].

Several studies have investigated the quality characteristics of tomato fruits ripened on-vine and off-vine [44,45]. However, limited information is available in relation to the postharvest storage ability of off-vine-ripened tomatoes for more than a few weeks or on the colour development of the fruit during the ripening process in relation to nitrogen supply.

Inspired by the foregoing considerations, the main aim of this study was to investigate the effect of nitrogen fertilization on the yield and quality characteristics of greenhouse tomatoes, aiming to improve advice about sustainable production for tomato growers in the Mediterranean area. Furthermore, we investigated the effect of nitrogen supply on unexplored but very important parameters, such as colour development during postharvest conditions, after complete maturation and shelf life of tomato, and provided valuable information about the physiology of the fruit during storage and at different stages of ripening and maturation.

## 2. Results

### 2.1. Yield Response

High nitrogen application significantly reduced the yield in Year 1 (2008) compared to the other nitrogen doses (*p* ≤ 0.001); however, nitrogen inputs did not significantly affect yield in Year 2 (2009) (Figure 1). Plants grown in Year 1 exhibited a higher yield compared with Year 2.

### 2.2. Time Course of Ripening and Maturation

High nitrogen application significantly reduced the duration from the breaker to the red stage for fruits ripened on the plants compared with standard nitrogen levels (*p* = 0.032), but the effect was not significant compared to low nitrogen application (Table 1). Fruit maturation was accelerated in Year 1 compared to Year 2; the number of days from breaker to red in Year 1 was significantly lower than those in Year 2 (*p* ≤ 0.001). Nitrogen input had no significant effect on the number of days from the breaker stage to the red stage in postharvest conditions.

There were no significant effects of the nitrogen inputs on the number of days from the flowering stage until entering the breaker and red stages, or from transplantation until entering the breaker and red stages (Table 1).

Higher average temperatures were recorded in Year 2 during June and July compared with average temperatures recorded in Year 1 during May and June (Table 2). In Year 2, plants flowered and reached the breaker and red stages during June and July, while plants grown in Year 2 flowered and reached the breaker and red stages during May and June.

### 2.3. Postharvest Storage Ability and Colour Development until Complete Ripeness

Low nitrogen applications significantly (*p* > 0.001) decreased the colour index a* compared with standard and high nitrogen doses during postharvest storage (Figure 2). The data were also analyzed to examine the overall effect of the treatments during the different days using repeated measures analysis. The overall effect of nitrogen on the colour index a* was found to be significant (*p* < 0.001).

Survival analysis showed no significant effect of the treatments (*p* > 0.200). The experiment was terminated after 147 days, even though eight fruits from high and standard nitrogen applications were still not rotten (Figure 3). Approximately 25 days after the fruit were harvested, the colour index a* stopped increasing and remained stable until the termination of the experiment (Figure 2).

### 2.4. Colour Indices

Tomato fruits in Year 1 exhibited significantly lower L*, a* and b* values compared with fruits in Year 2. The colour indices L* and b* were not significantly affected by the different nitrogen doses (Table 3). High nitrogen treatments significantly and consistently increased the colour index a* compared with low and standard nitrogen inputs (*p* ≤ 0.001).

### 2.5. Carotenoids

The effect of nitrogen supply on the total carotenoid content was not consistent during the experimental years (Figure 4). All colour indices were positively correlated with the total carotenoid content (*p* ≤ 0.003) (Figure 5). However, after adjustment for the effect of differences between years, the deviations of the colour index b* were no longer significantly correlated with the deviations in the total carotenoid content (*p* > 0.200), and the significance of L* was reduced (*p* = 0.038) (Figure 6).

## 3. Discussion

### 3.1. Yield Response

The effect of nitrogen treatments on the yield of tomato fruits was not consistent between the years, and environmental factors may have been the main reason for the lack of this consistency across years. A review reported that the effects of recommended nitrogen rates for tomato fruits varied significantly depending on experimental purposes and environmental factors [42], an observation that is in agreement with our findings. Plants grown in Year 1 exhibited a higher yield compared with Year 2. Many authors have reported that although deviations from the recommended nitrogen levels may not result in a significant difference in yield, climatic conditions (e.g., air temperature, rainfall pattern) significantly affect the yield of tomato fruits [22,46,47,48,49]. There is a threshold for nitrogen uptake at certain levels, which if exceeded results in decreasing yield. In general, it is assumed that overfertilization of greenhouse tomatoes will provide maximum production, and it has been established as widespread standard practice; however, it has negative environmental effects because it reduces the efficiency of fertilizers [50,51]. Frías-Moreno et al. [52] reported that the yield, weight and diameter of fresh tomato fruits increased as nitrogen supply increased up to an optimum nitrogen level (30 mmol∙L^−1^) beyond which growth attributes were reduced. According to a previous study [39], although an increase in nitrogen dose up to a certain level enhances yield, excessive nitrogen has an overall negative effect. Parisi et al. [53] reported that high nitrogen supply (150–250 kg nitrogen ha^−1^) caused no significant increase in the total ripe and unripe yield of tomato. Elia and Conversa [54] evaluated four nitrogen fertilization rates (1, 100, 200 and 300 kg∙ha^−1^) and showed that the maximum total yield was obtained with 200 kg∙nitrogen ha^−1^. Ozores-Hampton et al. [55] showed that maximum marketable yield was achieved by applying nitrogen doses between 172 kg∙ha^−1^ and 298 kg∙ha^−1^, whereas the application of 470 kg∙ha^−1^ of nitrogen considerably decreased yield. Li et al. [56] stated that higher nitrogen inputs > 300 kg ha^−1^ did not result in higher yields. Gent [57] concluded that an excess supply of nitrogen or potassium in hydroponic solution had little effect on the yield and fruit quality of greenhouse tomatoes. In addition, Bernard et al. [22] reported that lowering nitrogen supply from 12 to 6 or 4 mM NO_3_^−^ had a minimal impact on the commercial yield (7.5%) and quality of tomato fruit during the growing season. In general, optimal nitrogen supply could result in a higher yield when compared with no nitrogen supply [42].

In our study, there were no significant differences between the ‘Low Ν’ and ‘Standard Ν’ treatments, while the negative effects of the ‘High Ν’ treatment were not consistent between the years. Therefore, our recommendation would be to apply the lowest nitrogen dose, this would help to minimize the environmental impact without reducing or jeopardizing yield or quality. These suggestions are supported by Albornoz [58], who concluded that nitrogen management is essential for achieving maximum marketable yields and promoting environmental sustainability.

### 3.2. Time Course of Ripening and Maturation

Application of high nitrogen had a significant and negative effect on the duration from the breaker stage to the red stage for fruits ripened on the plants. Parisi et al. [53] reported that excess nitrogen supply (250 kg nitrogen) resulted in less concentrated ripeness and deterioration of the phytosanitary state (increase in the incidence of viral damages) and of some fruit attributes (soluble solids, glucose, and fructose content, reducing sugar/total solids ratio, pH). However, in our study, the effect of nitrogen during the years was not consistent, and fruit maturation was accelerated in Year 1, compared to Year 2. The ripening process was delayed by 40% for fruits that matured in postharvest conditions compared with fruits that matured on the vine. This effect could be due to the exposure of fruits that are growing in the greenhouse to sunlight, causing ripening that is accompanied by a peak in respiration and a concomitant burst of ethylene [44]. Such biological reactions generally cause a two- to three-fold increase for every 10 °C rise in temperature within the range of temperatures normally encountered in the distribution and marketing chain [59]. The postharvest temperature in the present study was similar to the minimum (night) temperature in the greenhouse during this period (Table 2) and much lower than the maximum day temperature.

The effect of nitrogen inputs on the plants’ duration from the breaker stage to the red stage was statistically significant (*p* = 0.032); however, under postharvest measurements, the effect was negligible (*p* = 0.457). Tomato sampling and greenhouse measurements were taken from the first fruit of the first truss; therefore, the effect of nitrogen in the greenhouse was direct because plants were at the early stages of productivity and the fruit load was low. This evidence supports the idea that climatic conditions may play a determinant role in the ripening process.

No significant effects of the nitrogen inputs were found on the number of days from the flowering stage/transplantation stage until entering the breaker and red stages, or from transplantation until entering the breaker and red stages. In Year 1 and Year 2, plants flowered 20 days and 27 days after transplantation, respectively. Higher average temperatures were recorded in Year 2 during June and July, compared with average temperatures recorded in Year 1 during May and June. Dorais et al. [60] reported that temperatures below 10 °C and above 30 °C inhibited the lycopene biosynthesis, which is the carotenoid responsible for the red colour of tomatoes, and suboptimal temperatures slowed down the ripening process of tomato fruits [61]. Thus, high temperatures can accelerate the maturation; however, they can delay the ripening process. Plants in Year 2 required more days to enter the breaker stage compared to plants grown in Year 1, while in Year 2, the ripening process (breaker stage–red stage) was accelerated.

### 3.3. Postharvest Storage Ability and Colour Development until Complete Ripeness

Application of a low nitrogen dose significantly decreased the colour index a* (redness of hue) compared with the rest of the nitrogen doses during postharvest storage. Massantini et al. [43] recorded an increase in the values of L* (lightness of colour intensity), b* (yellowness of hue) and a*, as well as a decrease in the a*/b* ratio with increasing nitrogen dose. The overall effect of nitrogen on the colour index a* was found to be significant. Huff [62] reported that discontinuation of the nitrogen supply reduced the re-green tendency of the pericarp of citrus fruits. Effects of nitrogen doses on the chlorophyll synthesis of plants were also reported in [63]. Organic tomatoes needed a greater number of days to reach the same values of the colour index a* indicating a delay in maturity compared with conventional fruits [64]. The above findings agree with the current study; this indicates that high nitrogen doses may affect the synthesis of plastids at the green stage and the conversion of chloroplasts to chromoplasts. Thus, fertilization applications influence the ripening of fruits, and high levels may accelerate the process. However, in the present study, since this was only tested in one of the two years and the effect was smaller than the changes in colour saturation that occurred during the postharvest period, this effect was not considered to be sufficiently important to affect our recommendation regarding nitrogen supply.

The postharvest experiment was performed at room temperature, without exposure to refrigeration since refrigeration conditions are known to negatively affect the quality of tomatoes [65]. Farneti et al. [66] found that a 4 °C temperature stimulates firmness and decay, increases fruit susceptibility to mechanical injury, decreases sugar and increases acid content when compared to fruit stored at 15 °C. In our study, when tomatoes were kept at a constant room temperature after harvest, they exhibited an unexpectedly extended storage period, averaging 93 days. Survival analysis showed no significant effect (*p* > 0.200) of any of the nitrogen applications on the storage duration of tomato fruits, from the mature green stage until rotting. After harvest and 25 days later, the colour index a* stopped increasing and remained stable until the termination of the experiment. This indicates that carotenoid synthesis continued for some time during the red stage, and subsequently, the carotenoid concentration of tomato fruits remained stable.

### 3.4. Colour Indices

In the present two-year study, colour indices L* and b* were not significantly affected by nitrogen input. A high nitrogen dose significantly increased the colour index a* compared with low and standard nitrogen inputs; this observation is in agreement with other studies [37,43,67]. In Year 2, fruits exhibited higher a* values than in Year 1. These values were five-fold greater than the effect of nitrogen supply; this could be due to different environmental conditions during the growing periods or during the harvests. Variations in colour development between different experimental years have also been previously reported [68,69], which were attributed to different climatic conditions that may have affected the colour of the surface mainly due to high light intensity reducing the chlorophyll content and promoting the synthesis of carotenoids [70]. Zhou et al. [71] concluded that tomato plants developed their defense systems, including chlorophyll loss and the synthesis of carotenoids, to protect themselves from multiple stress factors, such as high light intensity and heat stress.

### 3.5. Carotenoids

The economic cost of fertilizers is increasing, optimizing the supply levels to obtain high yield without compromising the quality could decrease the use of fertilisers and reduce the cost. Nitrogen fertilizers generally increase the carotene concentrations in plants [31]. However, these results were not consistently observed in the present study (Figure 4), supporting the suggestion that most of the treatments provided a super-optimal supply of nitrogen. Nitrogen fertilization may considerably affect various quality attributes of tomato fruits [72,73]. Dumas et al. [13] reported that the lycopene concentration in tomatoes increases when the nitrogen supplied to the crop decreases, whereas the yield increased with the highest nitrogen level. In contrast, small increases in nitrogen fertilization during cultivation increased the lycopene content of the fruit [74,75]. San Martín-Hernández et al. [73] concluded that, when nitrogen is provided at suboptimal levels, lycopene content is positively affected, whereas when nitrogen is supplied at sufficient or elevated levels, quality attributes are negatively affected [76,77].

Colour indices were positively correlated with the total carotenoid content. Bénard et al. [22] observed that carotenoid concentration did not respond significantly to nitrogen doses and reported that other factors, such as fruit irradiance and temperature fluctuations, played important roles in the carotenoid composition of tomato fruits; these observations agree with the current study.

According to the results of this study, the colour index a* and the hue angle appear to be good indicators of the total carotenoid content of tomato at the red stage. Massantini et al. [43] suggested that changes in the values of L*, a* and a*/b* could be associated with lycopene synthesis. Pandurangaiah et al. [78] recorded strong positive correlations between the colour index a* and total carotenoids (R^2^ = 0.82) as well as a positive correlation between the colour index a* and lycopene content (R^2^ = 0.87) of tomato in the ripe stage; the hue angle also showed a strong positive correlation with β-carotene content. The authors suggested that the close association between colour and carotenoids, which was established through colorimeter readings, could be a suitable method to improve the nutritional value of tomatoes. Similar results have been observed in other studies that [40,79] show correlations that rely on the physiological ripening process of the fruits rather than on the responses of the colour and the carotenoid concentration after complete ripeness. The above-mentioned study also investigated correlations between colour indices and carotenoid content at different maturity stages (ripening at green, breaker, turner and ripe stages). In one study, the L*, a*, b*, hue, chroma and lycopene content were plotted against the various maturity stages of fruit [40], and a correlation was observed between the L* and lycopene content. In addition, an increase in the lycopene concentration resulted in a corresponding decrease in L*, altering the colour of the tomato from light to dark. The authors also reported that the increase in the a* and a*/b* ratio was related to lycopene synthesis. In addition, Kaur et al. [79] found a significant increase in lycopene content during ripening in all cultivars. Cox et al. [80] reported that yellow, orange, and red fruit colour correlates to lycopene content and that lycopene content in fruit is influenced by photoperiod during ripening. San Martín-Hernández et al. [73] stated that the concentration of carotenoids and the nutritional value of the tomato fruit were influenced by nitrogen and potassium nutrition at phenological stages. The lycopene content is a good index of maturation level [81,82] because as tomatoes develop and mature from green to ripe, the increase in carotenoid content is closely related to the increase in the lycopene content.

## 4. Materials and Methods

### 4.1. Experimental Design and Management

The experiment was conducted with tomato (*Lycopersicon esculentum* Mill.) plants in a plastic greenhouse without heating at the farm of the Department of Soil Science of Athens, Institute of Soil and Water Resources, Hellenic Agricultural Organization—DIMITRA, in the Attika region, Greece. The experiment was replicated for 2 years to support the general applicability of the results. It took place during the spring season between April and July of 2008 and 2009 (Year 1 and Year 2). The experiment was a fully factorial randomized design laid out in three blocks using the tomato variety Elpida (Enza Zaden), an indeterminate type for fresh consumption. In each block, nine tomato plants per treatment were fertigated at three nitrogen levels (low = 6.4 mM/plant, standard = 12.9 mM/plant, high = 25.7 mM/plant). The standard nitrogen dose, throughout each experimental period, corresponded to a total of 4.32 g nitrogen/plant or 130 kg nitrogen/ha considering plant populations of around 30,000 plants/ha.

The greenhouse experiment was conducted according to usual Greek agricultural techniques and in consultation with Greek tomato growers. The seedlings were 12–17 cm high, had developed their third true leaf, and their biological age was approximately 40–45 days when they were transplanted into 35 cm diameter pots (20 lt). The pots were filled with a mixture of soil, compressed peat, and inorganic perlite (3:1:1). The soil was obtained from the surface layer (0–40 cm deep) of the DSSA farm. The soil was then ground, sieved, and analyzed before being used.

The greenhouse was sterilized with chlorine. Plants were stringed approximately 20 days after transplantation, and they were disbranched every 10 days to enhance their development. The greenhouse microclimate was not controlled. The abiotic conditions, minimum and maximum temperatures (°C), minimum and maximum relative humidity (RH%), soil temperature during morning and afternoon hours and light intensity (expressed as μmol m^−2^ s^−1^) during afternoon hours were recorded every day (Table 2). Minimum and maximum temperatures and RH% were used to calculate the vapour pressure deficit (VPD). The plants received water through a drip irrigation system and nitrogen three times per week through watering. The everyday water quantity (mL/day) and duration (min/day) were adjusted according to the environmental conditions and the plant’s developmental stage.

### 4.2. Soil Sampling and Chemical Analysis

The preparation of samples for analysis was performed according to the ISO 11464:2006 method (Soil quality—pretreatment of samples for physico-chemical analysis). Laboratory determinations were performed according to methods routinely used for soil characterization [83]. Particle-size distribution was determined by the Bouyoucos method; pH and EC were measured in paste extract with a pH/EC meter equipped with a glass electrode; carbonates were measured by using a Bernard calcimeter; soil organic C was determined by sulfochromic oxidation [84]; total nitrogen by the Kjeldahl method [85]; available phosphorous by sodium hydrogen carbonate extraction [86]; exchangeable K, Ca and Mg by BaCl_2_ extraction [87]; and available Μn, Fe, Zn and Cu by DTPA extraction [88]. The soil properties are presented in Table 4.

### 4.3. Harvests and Data Recorded during the Experiments

Tomato fruits were harvested at different stages according to the USDA tomato ripeness colour chart [89]. For analytical purposes, four fruits from each plant were harvested twice during the growing period. The fruits were cut into small pieces and frozen at −20 °C; pooled samples consisted of one quarter from each tomato fruit.

The fresh weight of each fruit and yield of each plant were recorded individually. Days from the breaker stage to the red stage were recorded for the first fruit of the first truss without removing the fruits from the plants. Days from transplantation and flowering until the breaker and red stages were recorded for the first flower and the first fruit of the first truss. In Year 2, fruits were removed from the plants at the breaker stage, and the days until they reached the red stage were recorded without refrigeration. Additionally, to test the postharvest storage ability, another batch of tomatoes was harvested at the mature green stage and fruits were kept at room temperature (21 ± 1 °C) without refrigeration. The number of days until the fruits began to rot was recorded, and the colour indices (L*, a*, b*) were measured regularly during the initial 45 days of the postharvest period.

### 4.4. Carotenoid Extraction and Analysis

#### 4.4.1. Extraction

Freeze-dried tomato samples were extracted and processed based on previously described methods [90]. Samples of 1 g were homogenized with an Ultra Turrax homogenizer (T25 basic, IKA Werke GmbH & Co. KG, Staufen, Germany) by adding 15 mL of ethyl acetate (EtOAc) and stored overnight at 4 °C. The supernatant was removed after centrifugation at 4000 rpm for 10 min. The residues were re-suspended twice in 2–3 mL of EtOAc with overnight extraction for another two days following the same procedure described above. The supernatants that were collected during the three-day extraction procedure were combined, centrifuged and filtered through a Whatman No 1 filter paper. Twenty μL were subjected to high-performance liquid chromatography analysis (HPLC).

#### 4.4.2. HPLC Method

The chromatographic apparatus consisted of two Gilson pumps (302, 305), a mixer (805 manumetric module, maximum pressure 60 MPa Gilson), an injector (234 sampling injector Gilson), a column heater purchased from Jones chromatography and a UV-Vis detector (Gilson Holochrome, 200–600 nm). The column was a Phenomenex Hyper Clone 5 μ BDS C18, 1 (250 × 4.60 mm), 5 micron column. A Waters In-Line Spherisorb ODS2 guard column was used. For data processing and analysis, the Shimadzu GC solution version 2.3 software was used. The mobile phase consisted of (A) methanol (MeOH) at 100% and (B) EtOAc:MeOH (50:50 *v*/*v*). The mobile phase was filtered through a 45 μm membrane and degassed ultrasonically prior to use. The solvents were obtained from Fisher scientific or HiperSolv CHROMANORM and were of HPLC grade. The gradient was 0–8 min 100% A, 33 min 30:70% (A:B), 33–35 min 30:70% (A:B), 37–42 min 100% B and 42–50 min 100% A. The flow rate was 0.8 mL/min, and the column temperature was set at 30 °C. The absorbance was read at 450 nm. The retention times for each compound were as follows: lutein 7.2 min, unidentified compounds 19.5, 20.1, 21.5, 27.0 min, all-trans lycopene 27.8 min, 9 cis lycopene 29.1 min and β-carotene 30.5 min. The peak spectra were checked with a diode array detector (DAD). The samples were quantified by comparing peak areas with a lycopene standard purified in the laboratory; commercial β-carotene was obtained from Sigma (approximately 95% UV), and a lutein standard was kindly provided by Dr. Georg Lietz (Human Nutrition Research Centre, Newcastle University, UK).

#### 4.4.3. Lycopene Standard Preparation and Purification for HPLC Quantification

Fresh tomatoes were cut into pieces after removing the seeds and then frozen. The frozen material was freeze-dried. EtOAc (500 mL) was used to dissolve pigments. The extract was left overnight at room temperature, protected from light and 0.70g of anhydrous sulphate (Na_2_SO_4_) was added to remove water. Subsequently, the extract was filtered through Whatman No 1 filter paper and evaporated to dryness in a rotary evaporator (26 °C). The residue that was obtained was dissolved in 10 mL of hexane, centrifuged at 4000 rpm for four minutes, and the supernatant was collected and chromatographed on a silica gel column (Kieselgel type 230–400, grade 9385 mesh, 60 Aº pore diameter) 45 cm in length with a 2.5 cm inner diameter and the separation of the different pigments was achieved by adding different mixtures of hexane/EtOAc in the following order: 90:10, 80:20, 70:30, 60:40, 50:50, 40:60, 30:70 and 20:80 (*v*/*v*). The resulting fractions were further analyzed by HPLC.

The HPLC apparatus and the mobile phase were the same as described above. The gradient was (A) ACN: Water (70:30 *v*/*v*), (B) EtOAc: ACN (50:50 *v*/*v*), 0–25 min 100% A, 100 min 50:50 (A:B), 120 min 100% B, 150 min 100% B and 160–180 min 100% A. The column was a Phenomenex Luna 5 μ C18 (2) 100A (250 × 21.20 mm), at 5 microns. The flow rate was 3 mL/min. The fractions from the silica gel column were separated by introducing the sample via the solvent. Coloured fractions were collected every 30 s. All-trans lycopene occurred at 120 min. Furthermore, the fraction purity was evaluated by injecting the fractions separately to the HPLC with an analytical column. The absorption of the lycopene fraction was measured spectrophotometrically with a Shimadzu spectrophotometer (UV-240 1PC) to determine the concentration. Lycopene was obtained with a purity of 95–96% (HPLC at 450 nm).

### 4.5. Colour Indices

Colour indices (L*, a*, b*) were measured with a Minolta chromameter (CR-200). Each fruit was measured individually along its equator. The hue angle (a/b*) was calculated by dividing colour index a* by colour index b* only for the correlations between the colour indices and the carotenoid content in red tomato.

### 4.6. Statistical Analysis

All the data were subjected to statistical analysis using the general linear model (GLM) in Minitab 15. In case the estimated values of the mean were closer to the sample median after logarithmic transformation than before and the residuals were normally distributed, the back transformed means were PCA calculated and expressed on a logarithmic scale. In order to adjust for the year effect on the correlated parameters, the deviations from the yearly averages were calculated as described in [91]. The repeated measures analysis was conducted with SAS 92.

## 5. Conclusions

The different treatments had no consistent effects on the yield across the years. Environmental conditions played a key role in the maturation and ripening process of the fruits the maturation process was delayed by harvest at the breaker stage (2.5 days, *p* ≤ 0.001) or by super-optimal temperatures in the second year of experimentation (10 days, *p* ≤ 0.001). The tomato variety Elpida exhibited a prolonged postharvest storage period when not exposed to refrigeration. High nitrogen treatments accelerated the ripening process during postharvest storage and caused a significant increase in the colour index a*. The sum of total carotenoids exhibited positive correlation with L*, a* and the hue angle (a/b*); however, differences in b* were more dependent on the year of cultivation. There were no consistent effects between the years on the total carotenoid content or in the yield or quality. Based on the results of this study, it would be more beneficial to supply the lowest nitrogen doses. Determining optimal levels of nitrogen for certain crops could minimize the environmental impact caused by excessive use of fertilizers and promote crop sustainability through the effective and efficient management of plant nutrients without jeopardizing quality for consumers or profit for growers. Further research to investigate the postharvest storage ability of different tomato varieties and to correlate these observations with other organoleptic and nutrient information is recommended.

## Figures and Tables

**Figure 1 plants-12-01553-f001:**
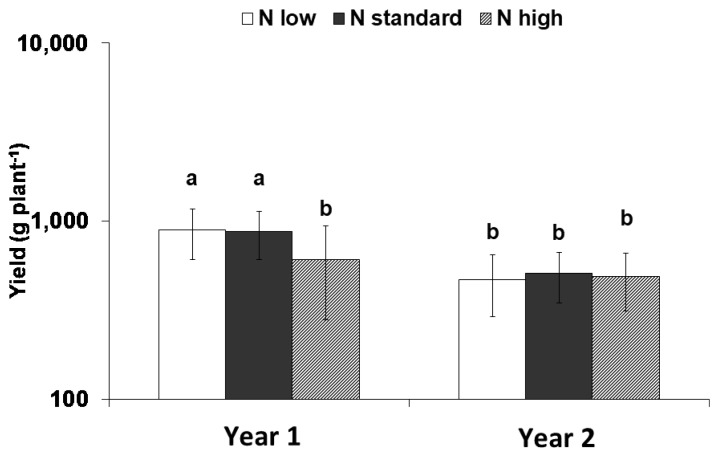
Interaction between nitrogen input and the experimental years on the yield. Means with different letters are significantly different (*p* < 0.05). Error bars indicate standard deviations. Means are expressed on a logarithmic scale.

**Figure 2 plants-12-01553-f002:**
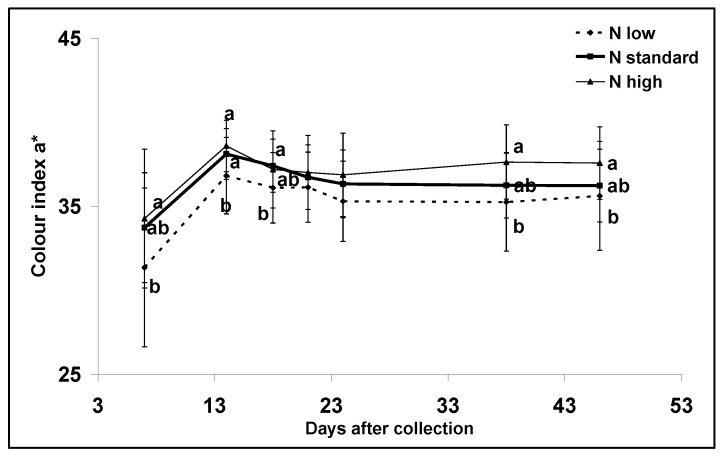
Nitrogen input effect on colour index a* during postharvest storage from the pink stage to complete ripeness. Means with different letters are significantly different (*p* < 0.05). Error bars indicate standard deviations.

**Figure 3 plants-12-01553-f003:**
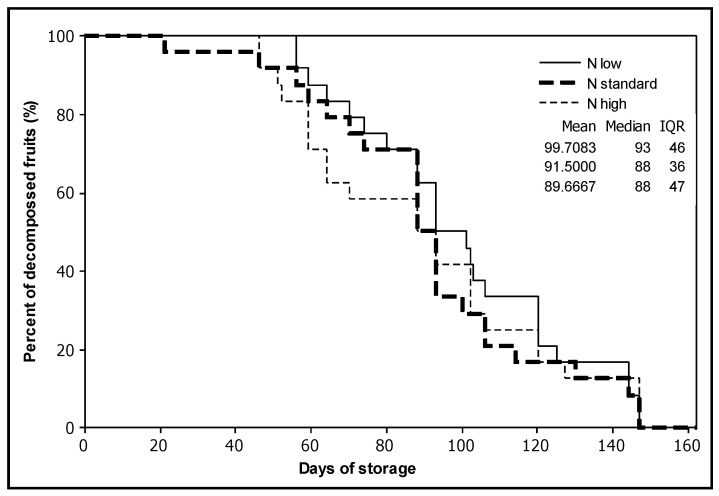
Survival plot for the effect of the nitrogen treatments on the number of days of storage at room temperature until tomato fruits began to show signs of rotting (the test was terminated at 147 days).

**Figure 4 plants-12-01553-f004:**
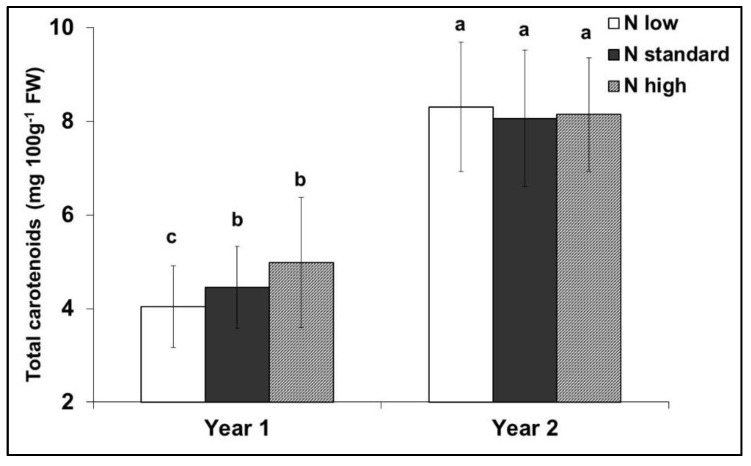
Interaction between nitrogen input and the experimental years on the total carotenoid content of red tomato fruits. Means with different letters are significantly different (*p* < 0.05). Error bars indicate standard deviations.

**Figure 5 plants-12-01553-f005:**
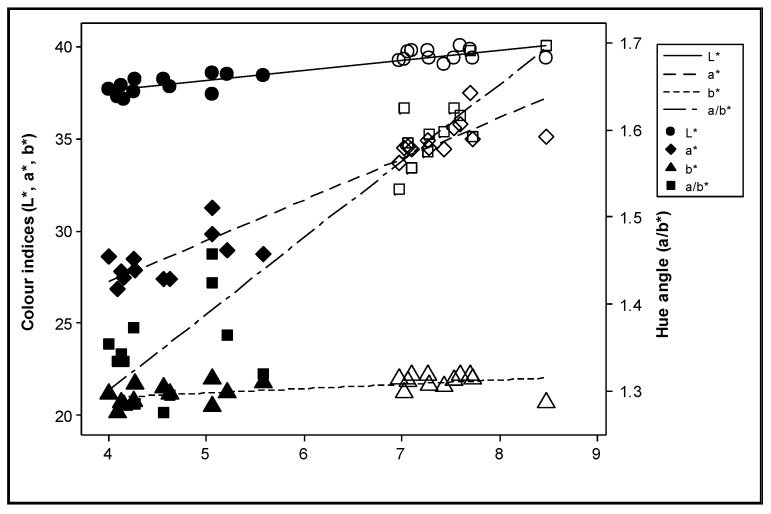
Correlation between the colour indices L* (lightness of colour), a* (redness of hue), b* (yellowness of hue) and a/b* with the sum of all-trans lycopene, 9 cis lycopene and β-carotene during the experimental years. Solid symbols represent values in Year 1, while open symbols represent values in Year 2.

**Figure 6 plants-12-01553-f006:**
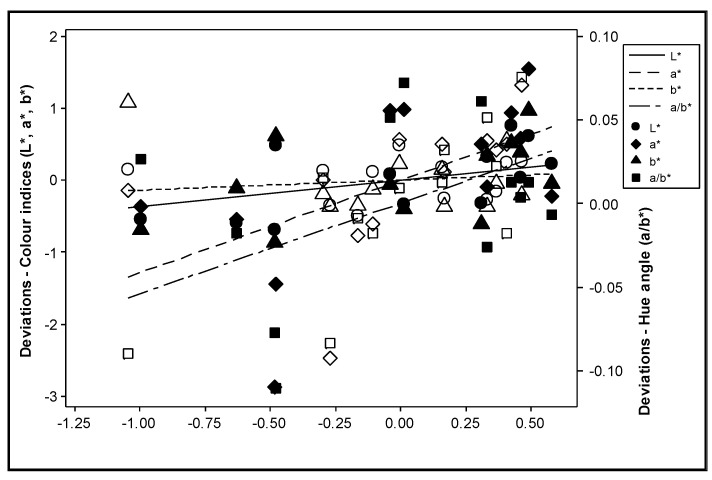
Correlation between the deviations from the year means for the colour indices L*, a*, b* and a/b* and the deviations from the year means for the sum of all-trans lycopene, 9 cis lycopene and β-carotene during the experimental years. Mean values = the average values of the data in each year. Deviation = the measured values minus the corresponding mean values. Solid symbols represent values in Year 1, while open symbols represent values in Year 2.

**Table 1 plants-12-01553-t001:** Analysis of variance (ANOVA) of the number of days from the breaker stage to the red stage both on the plants and at postharvest conditions, and of the days from flowering and transplantation until the breaker and red stages in Years 1–2.

	Breaker to Red Stage					
Factors	On the Plants	Postharvest Conditions	Flowering to Breaker Stage	Flowering to Red Stage	Transplantation to Breaker Stage	Transplantation to Red Stage
**Nitrogen input**						
low	3.9 ±1.1 ab	6.0 ± 1.8	42.4 ± 1.8	46.4 ± 0.7	66.7 ± 1.5	70.7 ± 0.5
standard	4.0 ± 1.3 a	6.5 ± 1.5	43.1 ± 0.7	47.1 ± 1.7	67.6 ± 0.4	71.6 ± 1.5
high	3.7 ± 1.0 b	6.9 ± 1.7 ns	41.2 ± 0.8 ns	44.9 ± 0.8 ns	66.1 ± 0.5 ns	69.8 ± 0.5 ns
**Year**						
Year 1	3.0 ± 0.7 b		49.3 ± 1.3 a	52.4 ± 0.3 a	69.3 ± 2.3 a	72.4 ± 0.3 a
Year 2	4.3 ± 1.1 a		38.4 ± 2.3 b	42.8 ± 1.4 b	65.4 ± 2.3 b	69.8 ± 1.4 b
**ANOVA *p*-values**						
Nitrogen input (N)	0.032	0.457	0.210	0.151	0.210	0.151
Year (Y)	≤0.001		≤0.001	≤0.001	≤0.001	≤0.001
Days (D)		≤0.001				
N × Y	0.412		0.675	0.551	0.675	0.551

Mean values (±standard deviations) within the same factor followed by different letters are significantly different. ns: no significant differences.

**Table 2 plants-12-01553-t002:** Abiotic conditions in the greenhouse during the experimental years.

Abiotic Conditions in the Greenhouse						
Year	Month	T (°C) (min/max)	RH% (min/max)	VPD (kPa)	Soil T (°C) (Morning/Afternoon)	μmol m^−2^ s^−1^ (Afternoon)
**Year 1**	**April**	12.4/27.9	24/72.4	1.22	18/23.4	1.35 × 10^3^
	**May**	13.9/31.2	24/68.2	1.47	20.5/23.9	1.29 × 10^3^
	**June**	19.2/34.6	24/60.5	2.04	23.5/26.3	1.62 × 10^3^
	**July**	21.5/36.5	24/49.5	2.53	25.6/29.5	-
**Year 2**	**May**	14.8/32.2	24.3/64.5	1.61	20.2/25.6	1.74 × 10^3^
	**June**	19.1/37	24/59.3	2.21	23.9/27.6	1.74 × 10^3^
	**July**	21.2/38.6	24/60	2.44	25.2/29	1.65 × 10^3^

**Table 3 plants-12-01553-t003:** Analysis of variance (ANOVA) for the colour indices (L*, a*, b*) in Years 1 and 2.

	Colour Indices (L*, a*, b*)		
Factors			
**Nitrogen input**	L*	a*	b*
low	38.44 ± 1.99	30.36 ± 4.07 b	21.23 ± 2.32
standard	38.47 ± 1.65	30.82 ± 4.14 b	21.27 ± 2.12
high	38.69 ± 1.61 ns	31.88 ± 4.20 a	21.54 ± 2.03 ns
**Year**			
Year 1	37.91 ± 1.63 b	28.51 ± 2.80 b	21.11 (±1.95) b
Year 2	39.51 ± 1.50 a	34.97 ± 2.66 a	21.72 (±2.42) a
**ANOVA *p*-values**			
Nitrogen input (N)	0.490	≤0.001	0.317
Year (Y)	≤0.001	≤0.001	0.001
N × Y	0.272	0.650	0.835

Μean values (±standard deviations) within the same factor followed by different letters are significantly different. ns: no significant differences.

**Table 4 plants-12-01553-t004:** Main soil properties.

Texture	pH	E.C.	Org. Matter	CaCO_3_	Nitrogen	P-Olsen	Exchangeable Cations			DTPA Extractable Micronutrients			
							K	Mg	Ca	Zn	Mn	Fe	Cu
		μS/cm	g kg^−1^	%	g kg^−1^	mg kg^−1^	cmol kg^−1^			mg kg^−1^			
SCL	7.75	489	17.1	29.05	1.05	10.5	0.88	1.10	10.51	2.76	6.75	5.9	3.47

## Data Availability

The data presented in this study are available from the authors upon request.

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
