# Peer review of "Nitrogen Application Can Be Reduced without Affecting Carotenoid Content, Maturation, Shelf Life and Yield of Greenhouse Tomatoes"

_plants, 2023, doi:10.3390/plants12071553_

Round 1

Reviewer 1 Report

1. The plant material used for sample extraction has to be described more precisely, in the manuscript there is no information about how many plants were planted or sampled in the experiments.

2. The authors discussed that different Nitrogen application had no consistent effects on the yield and the most important factor that affected the quality characteristics was the climatic conditions. Data showed that average temperatures and light intensity were recorded in two years of experiments. However, the author did not discuss the effect of light intensity on tomato fruit yield and colour indices or carotenoid content. The light conditions are closely related to the biosynthesis of plant pigments and fruit colour. Hope the authors can supplement this discussion.

Author Response

We would like to thank the reviewer for the valuable comments and we hope that our answers to the reviewer's comments and the revised manuscript will improve the quality of the manuscript. In addition, the English language and style have been checked by a native English speaker. Please see the attached file.

Reviewer 2 Report

The ms plants-2269598 with the title of Resilience of carotenoid content, colour indices, post and pre-harvest maturation, shelf life and yield of greenhouse tomatoes to changes in nitrogen supply investigates the effect of N fertilization on yield and quality characteristics of greenhouse tomato aiming to improve advice about sustainable production for tomato growers in the Mediterranean area. The ms need significant improvement to be acceptable for publication in such high quality journal.

The tile should be revised and the authors should make it shorter and more attractive instead of making it descriptive. For example, they can use one words to express different words of different traits.

L16: Based on what authors decided to investigate these three different N levels (low 6.4, standard 12.8, high 25.9 mmol/plant)?

L17: change (P<0.001) to (P0.001), check this issue in whole ms please

Keywords: add the Latin name of the tomato

L28-29 add the Latin name of tomato in the first sentence directly after the English this plant.

I prefer the authors use tomato instead of tomatoes in the whole ms.

L39: According to …? Add the name of the authors/author before the number of the ref in this case.

L38-39: please cite this ref:  https://doi.org/10.3390/su12062159

L45: please correct the superscripts and subscripts issues in the whole ms, such as NO3−!

L87: main hypothesis! Or  main aim? Please correct and add the hypothesis of these aims at the end of the introduction.

L100 High? Or The highest?

L344: If the experiment was conducted at plastic greenhouse, then why the authors investigated the year’s effects?

L347: season between (April-July) for two consecutive years (Year 1 and Year 2)? What were the years exactly? 2015? 2020? Or …?

L347-348: fully factorial randomized design? But what was the basic design for your experiment? Factorial in a completely randomized design (CRD) with or in a completely block design?

L350: (low=6.4 mM/plant, standard=12.9 mM/plant/, high=25.7 mM/plant), how many grams N per plant during the growth period? 4.32 g N/plant or 130 kg N/ha, why authors did not present their treatments as grams?

Table 4: No need for it, you have already mentioned this information in the text several times. Remove this Table, please

Please add the original citations for the used methods in your measurements.

In conclusion, please add some key results for the most important findings.

References: Please follow the format of the journals, there bold and italic in the list of references, check the format, please

The authors should present the weather conditions for the two growing seasons since they analyzed the year’s effects on investigated traits. I really do not understand why authors investigated the years’ effects in such an experiment!

In Table 1 and other Tables: The authors should add the SD after the mean as mean±SD instead of mean (SD). I am talking about presenting the values in the Tables.

I prefer the authors to use SD instead of SE in their Tables and Figures.

Regards, Reviewer

Author Response

(The authors gave the same response as above.)

Round 2

Reviewer 2 Report

The ms has been improved